# Effects of anti-malarial prophylaxes on maternal transfer of Immunoglobulin-G (IgG) and association to immunity against *Plasmodium falciparum* infections among children in a Ugandan birth cohort

Erick Jacob Okek[1,2]*, Moses Ocan[2,3], Sande James Obondo[1], Anthony Kiyimba[4], Emmanuel Arinaitwe[2,4], Joaniter Nankabirwa[2,4], Isaac Ssewanyana[4,5], Moses Robert Kamya[2,4]

1 Department of Immunology and Molecular Biology, College of Health Sciences, Makerere University, Kampala, Uganda, 2 Department of Medicine, Malaria Research Training Program, College of Health Sciences, Makerere University, Kampala, Uganda, 3 Department of Pharmacology & Therapeutics, College of Health Sciences, Makerere University, Kampala, Uganda, 4 San Francisco Infectious Disease Research Collaboration, Makerere University-University of California, Kampala, Uganda, 5 Central Public Health Laboratories, Ministry of Health, Kampala, Uganda

* okekerick@yahoo.com

## Abstract

### Background

The in-utero transfer of malaria specific IgG to the fetus in *Plasmodium falciparum* infected pregnant women potentially plays a role in provision of immune protection against malaria in the first birth year. However, the effect of Intermittent Prophylactic Treatment in Pregnancy (IPTp) and placental malaria on the extent of in-utero antibody transfer in malaria endemic regions like Uganda remain unknown. The aim of this study was thus to establish the effect of IPTp on in-utero transfer of malaria specific IgG to the fetus and the associated immune protection against malaria in the first birth year of children born to mothers who had *P. falciparum* infection during pregnancy in Uganda.

### Methods

We screened a total of 637 cord blood samples from a double blinded randomized clinical trial on Sulfadoxine-Pyrimethamine (SP) and Dihydroartemisinin-Piperaquine (DP) IPTp in a Ugandan birth cohort; study conducted from Busia, Eastern Uganda. Luminex assay was used to measure the cord levels of IgG sub-types (IgG1, IgG2, IgG3 and IgG4) against 15 different *P. falciparum* specific antigens, with tetanus toxoid (t.t) as a control antigen. Man-Whitney U test (non-parametric) in STATA (ver15) was used in statistical analysis of the samples. In addition, Multivariate cox regression analysis was used to determine the effect of maternal transfer of IgG on the incidence of malaria in the first birth year of children under study.

**Data Availability Statement:** All relevant data are within the paper and its Supporting Information files.

**Funding:** O.E-Research reported in this publication was supported by the Fogarty International Center of the National Institutes of Health under award number D43 TW 10526.The content is solely the responsibility of the authors and does not necessarily represent the official views of the National Institutes of Health.

**Competing interests:** All Authors have declared that no competing interests exists.

## Results

Mothers on SP expressed higher levels of cord IgG4 against erythrocyte binding antigens (EBA140, EBA175 and EBA181) ($p<0.05$). Placental malaria did not affect cord levels of IgG sub-types against selected *P. falciparum* specific antigens ($p>0.05$). Children who expressed higher levels (75th percentile) of total IgG against the six key *P. falciparum* antigens (Pf SEA, Rh4.2, AMA1, GLURP, Etramp5Ag1 and EBA 175) had higher risk of malaria in the first birth year; AHRs: 1.092, 95% CI: 1.02–1.17 (Rh4.2); 1.32, 95% CI: 1.00–1.74 (PfSEA); 1.21, 95%CI: 0.97–1.52 (Etramp5Ag1); 1.25, 95%CI: 0.98–1.60 (AMA1); 1.83, 95%CI: 1.15–2.93 (GLURP) (GLURP), and 1.35,; 95%CI: 1.03–1.78 (EBA175). Children born to mothers categorized as poorest had the highest risk of malaria infections in the first birth year (AHR: 1.79, 95% CI: 1.31–2.4). Children born to mothers who had malaria infections during gestation had higher risk of getting malaria in the first birth year (AHR 1.30; 95% CI: 0.97–1.7).

## Conclusion

Malaria prophylaxis in pregnant mothers using either DP or SP does not affect expression of antibodies against *P. falciparum* specific antigens in the cord blood. Poverty and malaria infections during pregnancy are key risk factors of malaria infections in the first birth year of growth of children. Antibodies against *P. falciparum* specific antigens does not protect against parasitemia and malaria infections in the first birth year of children born in malaria endemic areas.

## Introduction

Pregnancy greatly changes the physiology of a woman and increases vulnerability to diseases such as malaria by lowering the immune system [1]. Infection of the placenta can lead to severe adverse birth out comes such as low birth weight, still birth, miscarriage and premature birth [2, 3]. In Uganda where malaria is endemic, there is very low prevalence of the disease in children below six months much as they are considered to be vulnerable population to malaria [4]. It is recommended for pregnant women to take at least a dose of IPT per trimester of gestation. Among factors hypothesized for low malaria prevalence in the pediatric population are; fetal hemoglobin that has high affinity for oxygen thus consumes most of the essential gas making it unavailable for parasite metabolism; lactoferritin in breast milk that tightly binds iron making it unavailable for parasite biofilm formation and growth, and IgG transferred from placenta to cord blood which helps clear parasites [5, 6] using different mechanisms.

Mothers from malaria endemic areas pass on immune IgG to their children [7]. IgG1 and IgG3 have higher serum concentration compared to IgG2 and IgG4 [8]. IgG1 and IgG3 are cytophilic, perform opsonization, activate complement and antibody dependent cellular cyto-toxicity (ADCC), making them critical in development of immunity against malaria [9]. IgG2 and IgG4 are non-cytophilic; expression of more IgG2 has been associated with increased risk of malaria infection [10, 11] while IgG4 does immune modulation and limits malaria disease severity. High expression of IgG4 has been associated with repeated exposure to specific *P. falciparum* antigens. As a result, individuals in malaria-endemic areas express higher levels of IgG4 [12]. Cord blood of children born to mothers in malaria endemic regions has high levels of anti-MSP antibodies [13]. A study in Maputo province of Mozambique showed that

placental infection was associated with an increase in maternal levels of total IgG and IgM against a broad range of *P.falciparum* specific antigens [14, 15].

A study in Papua New Guinea randomized mothers into two groups of Sulfadoxine- Pyrimethamine-Azithromycin and SP-chloroquine as malaria prophylaxes and found no difference in cord antibody levels (total IgG) between the two groups [16]. No easily available study directly compared effects of Sulfadoxine-Pyrimethamine and Dihydroartesimin-Piperaquine on levels of IgG sub-types in cord blood. In a case controlled study of pregnant women on anti-malaria prophylaxis in Nigeria, cord blood of mothers on prophylaxis had lower levels of IgG antibodies compared to the negative controls [17].

In this study, we determined if use of either DP or SP prophylaxis has any effects on cord levels of IgG. We further examined if placental malaria has any effect on expression of IgG in cord blood. Using cox regression, we modelled if cord blood levels of total IgG against Rh4.2, PfSEA, Etramp5Ag1, AMA1, EBA175 and GLURP181 adjusted for maternal wealth category index, gestational malaria prevalence, gravidity and birth weight have any effects on the risk of malaria infections in the first birth year.

## Materials and methods

### Study participants

Blood samples for the Neonates were collected from Masafu General Hospital (the main Referral health facility within Busia District). Pregnant mothers in the first trimester were enrolled from across 40 (forty) villages located within 30 Kms radius away from Masafu Hospital. Participants were drawn from the Sub-counties of Buteba, Busitema, Masaba, Lunyo, Bulumbi Masafu and Lumino. Demographic information and clinical data of study participants were captured(Mother and baby pair) (Table 2).

### Participants and data collection

The samples for this study were obtained from a previous study done among pregnant mothers. Cord blood samples were screened to assess the levels of malaria specific antibodies. Details of participants' demographics, socio-economic status and clinical information is contained in a published study. The current study obtained samples archived from the primary study, demographic and clinical details of participants can be found in a previous study [18].

### Sample collection for this study

Collection of the cord blood by the primary study took place at the time of delivery. Blood collection was done by a midwife, medical officer or obstetrician who conducted delivery or cesarian section. The blood was then centrifuged to obtain serum, archived at appropriate temperature until analysis. Archival was done and random-access numbers generated by the computer to help in tracking the samples in the biobank.

### Laboratory procedures

**Preparation of buffer A.** Buffer A was prepared following a method previously reported in a study by *I.ssewanyana et al*. Briefly, 1L of phosphate buffer solution was measured and put in a conical flask after which 500μl of tween (*Sigma-Aldrich; USA*) was dispensed into the PBS to form PBS-tween solution. 5g of PVP powder *(Sigma-Aldrich; USA)* was weighed using a weighing balance in aluminum foil and added to the above solution after which equal amount of PVA powder was weighed and added to the mixture; proper agitation and mixing was done using a vortexer. 5mls of BSA (*Thermofisher Scientific;USA*) was pipetted and added to the

solution; finally, 0.2g of Sodium azide *(Sigma-Aldrich; USA)* was weighed under a biosafety hood in an aluminum foil, added to the solution, properly vortexed and mixed. The final solution was labelled with day of preparation and names of those who prepared.

**Preparation of buffer B.** A method by [19] was followed in the preparation of buffer B. Briefly, 1000mls of phosphate buffer solution (PBS) was transferred to a sterile glass tube. To this was added, 500ul of tween solution, 5g of polyvinyl alcohol (PVA) *(Sigma-Aldrich; USA)*, 5mls of bovine serum albumin solution (BSA), 0.2g Sodium azide and 3mls of lyophilized *E. coli* solution (*Avanti Polar lipids 100600C, Sigma-Aldrich-USA*)). The mixture was incubated overnight at 4˚C.

## Dilution of cord plasma and expression of *P. falciparum* antigens

693μls of buffer B was transferred into deep well plates using multi-channel pipette. 7μl plasma was then dispensed to the same well. The procedure was repeated for all plasma samples and mixed by shaking. The final sample dilution used was 1 in 100. Diluted samples were then incubated in buffer B over night at 4˚C to allow *E. coli* extract mop out anti-*E coli* antibodies to minimize any background response that may be due to *E. coli* protein contamination of the antigens. The *P. falciparum* antigens were expressed in *E. coli*.

### *Plasmodium falciparum* recombinants coupled beads

A total of 16 recombinant *P. falciparum* pre-erythrocytic, erythrocytic and infected RBCs antigens, Tetanus toxoid (t.t-non malaria control) (*Microcoat GmbH*, *Germany*) coupled on Luminex beads were used in this experiment. Antigens were batched in three groups; (i) infected red blood cells associated antigens, (ii) merozoite apical complex expressed proteins, and (iii) merozoite surface antigens (Table 1).

### Dilution of *P. falciparum* antigens coupled beads

A standard protocol was used to make the dilution [19]. Briefly, 5mls of buffer A was transferred to a falcon tube. To this, 8μl of each bead region was added and mixed by shaking. This was repeated for all the 16 coupled beads antigenic regions.

**Table 1. Summary of the *P. falciparum* Blood stage specific antigens.**

| Description | Antigen name |
|---|---|
| Heat Shock Protein 40, type II, Antigen | HSP40Ag1 |
| Early Transcribed membrane protein 5 | ETRAMP5Ag1 |
| Schizont Egress Antigen | SEA |
| Apical Membrane Antigen 1 | AMA-1 |
| Erythrocyte Binding Antigen-140 Region III-V | EBA140RIII-V |
| Erythrocyte Binding Antigen-175 Region III-V | EBA175RIII-V |
| Erythrocyte Binding Antigen-181 Region III-V | EBA181RIII-V |
| Reticulocyte Binding Protein homologue 4 | RH4.2 |
| Reticulocyte Binding Protein homologue 5 | RH5 |
| Glutamate Rich Protein R2 | GLURP RII |
| 19kDa fragment of MSP1 molecule | MSP1-19 |
| Merozoite Surface Protein 2, Dd2 allele | MSP2Dd2 |
| Merozoite Surface Protein 2, CH150/9 allele | MSP2CH150/9 |
| Tetanus Toxoid (Non-adsorbed) | TT |
| Reticulocyte Binding Protein Homologue 22030 | RH22030 |
| Circumsporozoite Protein | CSP |

## MagPix multiplex bead array assay

IgG sub-types 1–4 against 15 *P.falciparum* blood specific antigens and Tetanus Toxoid (TT) was measured in plasma diluted in buffer B. 50µl of bead suspension was added to each well (1,000 beads/region/well) of the 96 plate (Bio-plex pro-flat bottom). The plate was washed, placed on a magnetic block for 2 minutes and the supernatant was then poured off. The beads were washed twice with PBS tween buffer, laid on the magnetic separator for 2 minutes and supernatant poured off.

Using a multichannel pipette, 50µl of prepared plasma at a dilution of 1in1000 in buffer B and pooled hyper immune control (27 adults from malaria endemic areas) was added to the beads; two wells had PBS added in them as blanks. The plates were then put on a shaker at 600 rpm for 90 minutes after covering the plates with aluminum foils at room temperature. After incubation, the plates were placed on a magnetic separator for 2 minutes, supernatant poured off by a rapid inversion with a sharp shake followed by a gentle blot on a paper towel. The plates were then washed three times using 200ul wash buffer. 50µl of a secondary antibody specific for an IgG sub-type (IgG1, IgG2, IgG3 and IgG4) with concentration of 1in1000 for IgG1, IgG2 and IgG4 while secondary antibody against IgG3 was diluted at 1in 2000 in dilution buffer and added to each well containing bound primary antibody-antigen complex, blank wells, positive control wells and negative control wells. Incubation was done for 60 minutes in a shaker. The plate was put on a magnetic separator for 2 minutes, supernatant poured off and washed three times using wash buffer to remove unbound antibodies.

Using a multichannel pipette, 50µl of 1in 200 R-Phycoerythrin-conjugate AffinPure F (ab') goat anti-human IgG (*abcam,Boston,MA,USA*) diluted in buffer A was added. The plate was then incubated at room temperature in a shaker at 600 rpm for 60 minutes. After the incubation, the plate was then placed on a magnetic separator for 2 minutes and supernatant poured off, washed three times using wash buffer. 100µl of plain PBS was then added to the plates. Incubation was done at room temperature by putting it on a shaker at 600rpm for 30 minutes. The plates were read on a MagPix platform (Luminex Corp, USA), acquiring at least 50 beads/region/well. The results were expressed as median fluorescence intensity (MFI). The blank well MFI (background effects) was deducted from each well to determine the net MFI (IgG positive result)

## Luminex assay and quality assurance

Luminex assay was done following modified manufacturer's guidelines. Positive control plasma samples were obtained from 27 adults' resident in malaria areas with known episodes of malaria and thought to be hyper immune. Negative controls samples were serum samples obtained from whole blood of 6 Caucasians from the United Kingdom with no known history of exposure to *P. falciparum*. Dilution curves were developed using positive control serum samples. For each plate run, negative and positive controls would be included. To ensure quality of results, weekly calibration of MagPix machine was done, sample probe blown and sonicated every morning prior to use, routine stringent cleaning using bleach and sodium hydroxide done to avoid clogging of the probe. A plate water run was done at the end of the day's sample run and a plate would be re-prepared if more than 10 wells had the bead counts less than 50. Other quality assurance measures included preparing fresh reagents after every three days or if there was evidence of contamination

## Data analysis

All Net fluorescence Index (Net MFI) were log transformed prior to analysis. Part of the net MFI was also normalized to minimize effects of extreme values on mean. Graphical

representation using box plot was used to compare mean levels of cord IgG sub types against *P. falciparum* signature antigens of infants whose mothers were on DP versus those on SP. Man-Whitney U test in STATA (ver 15); a non -parametric test was used to compare means of IgG1, IgG2, IgG3 and IgG4 levels against signature *P.falciparum* antigens for DP group compared to SP group. A statistically significant difference in the means of the two groups was confirmed if the P-value was less or = 0.05. To evaluate effects of placental malaria on cord expression of IgG sub-types, Man Whitney U-test and box plots were used to compare the difference in the mean of cord IgG sub-class levels of infants whose mothers had placental malaria versus mothers who never had.

Only IgG specific to one antigen at a time was used in building the Cox regression model while adjusting for non-IgG covariates at each time. The Cox regression model was built using back ward elimination method. For each IgG, the levels were grouped into percentiles, that is $25^{th}$, $50^{th}$ and $75^{th}$. The effects of different levels of the IgG ($25^{th}$, $50^{th}$ and $75^{th}$ percentiles) on the incidence of malaria among study participants were assessed. The levels of the IgG at the $75^{th}$ percentile was found to consistently have higher Hazard ratios and significant confidence intervals. Therefore, the levels of the IgG at the $75^{th}$ percentile were used in building the Cox regression model. The P-values of the IgG variables in the Cox regression models were adjusted for Multiple testing using the Benjamin-Hoceberg correction of multiple testing in excel. The non-IgG covariates included were birth weight ($<2500$ grams, $>2500$grams), maternal malaria incidence (No malaria detected, and malaria detected), Maternal wealth index (poorest, middle, and least poor), gravidity category (1–3 and 4+). Cofounding was assessed in the Cox regression model and any variable that had the percentage difference of more than 10% between the crude and adjusted Hazard ratios was considered a cofounder. In the model building, the covariates that had p-value of $\leq 0.2$ in the bivariate analysis were included in the model.

## Ethical clearance

The primary study was approved by the Research and Ethics Committee, School of Biomedical Sciences, Makerere University College of Health Sciences; Uganda National Council for Science and Technology (SBS 114; S1 Appendix). Informed consent was obtained after explaining possible risks and benefits of the research to participants. They were requested to sign after understanding the contents before maternal and cord blood draws. The study sought for permission to archive participants' samples for future use in research; those who declined to this had their samples discarded while keeping for those who consented. This current study was also approved by the Makerere University School of Biomedical Sciences Research and Ethics Committee (SBS 1012; S2 Appendix)

## Results

### Characteristics of study participants

Over half, 51.8% of mothers screened positive for malaria at enrollment. Most, 51.3% of the children were female. Some, 8.2% of the newborn children had a low birth weight. The majority, 72%, of the pregnant mothers had *P. falciparum* infection during the gestation period. Most, 62% of the children had at least one malaria episode during the first birth year (Table 2).

### Effects of IPT arm on cord levels of IgG sub-types

Significant differences in the mean of the two groups for IgG1 was only noted with tetanus toxoid (t.t) ($p = 0.0349$) with mean value of cord levels of IgG1 against t.t being higher for children

**Table 2. Summary of key characteristics of study participants.**

| CHARACTERISTICS | VALUE/PROPORTION |
|---|---|
| Mothers' parasite prevalence at enrollment | Blood slide microscopy positive = 51.8% |
| Child gender | Females:328/640 |
| Mother's prophylaxis arm | Dihydroartesinin-piperaquine (DP) = 48.6% while the rest were on SP |
| Placental malaria prevalence | 42% were placental malaria positive by histology. |
| Parasite prevalence by blood slide microscopy during pregnancy | 72% had at least one positive slide by microscopy from enrollment to labor pain |
| Episodes of malaria during the child's first year of life | At least a single malaria episode = 62% |
| Child hospitalization status | Hospitalized at least once during the first birth year = 3% |
| Birth weight | 8.2% had low birth weight |
| Episodes of fever | Highest number of fever episodes in a child = 16 |
| Mortality | 12 children died during the study period |

born to mothers on DP than those on SP (Fig 1A). Similar observation was noted for IgG3 against tetanus toxoid (p = 0.0378), with DP group expressing significantly higher levels of IgG3 antibody. Children whose mothers were on SP expressed higher levels of IgG1 against different antigens compared to DP though not statistically significant. There were no significant differences observed between the median values of the two groups for both IgG2 and IgG3 response against different *P. falciparum* specific antigens (Fig 1B and 1C). Most differences were observed with IgG4 against Erythrocyte Binding Antigens (EBAs) and Merozoite Surface Proteins (MSPs) (Fig 1D). For EBA140, EBA175 and EBA181, the mean cord levels of mothers on SP were significantly higher than those on DP (EBA140; *P* = 0.0.0481, EBA175;

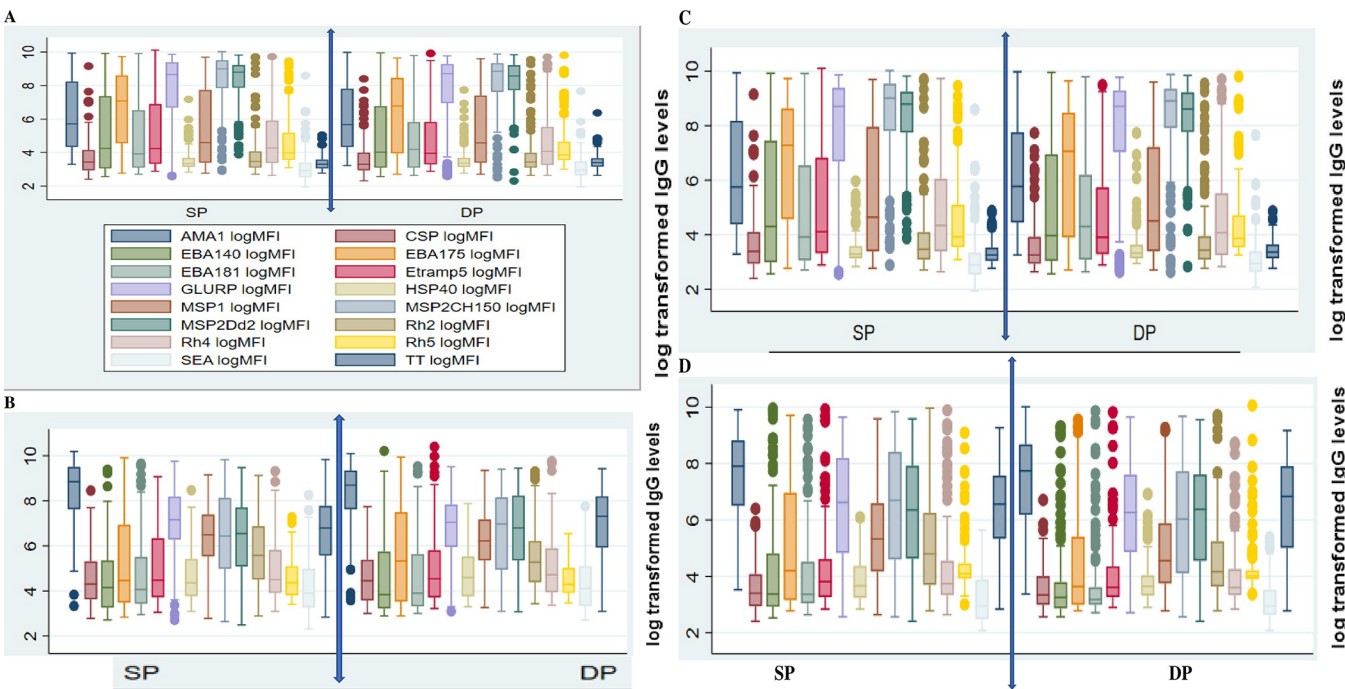

**Fig 1. Effects of mother's IPT arm on cord levels of IgG sub-types.** (A) comparison for IgG1; DP versus SP; (B) comparison of the two groups for IgG2; (C) effects of IPT use on IgG3 cord level; (D) comparison of the effects of the two groups of prophylaxis on IgG4 (all Y axes are in log MFI).

$P = 0.0170$, EBA181; $P = 0.0073$). Similar finding was noted with MSP2CH150 ($P = 0.0002$) and MSP2Dd2 ($P = 0.0073$) where mean levels of the SP group were significantly higher than the DP group.

## Association of placental malaria and malaria specific IgG antibodies

Placental malaria did not affect cord levels of different IgG sub-classes. This was done by comparing mean cord levels of mothers who were positive for placental malaria with the negative groups. The differences in cord levels of the various IgG sub-classes were evenly distributed among the two groups. For example, For IgG1 (Fig 2A) mean levels against GLURP is significantly higher for the negative group ($P = 0.0049$). For IgG3-(Fig 2B) mean levels for EBA140 and GLURP were significantly higher for the negative group ($P = 0.0448$ and $P = 0.0450$), while mean levels against HSP40 is higher for the placental malaria positive groups ($P = 0.0146$). For IgG2, (Fig 2C) mean levels against AMA1 is higher for the positive group ($P = 0.0125$) and the same for RH22030 (P = 0.0113). The same even distribution in significant difference was noted for IgG4 response against EBA140 where mean cord levels is higher for the placental malaria negative group (P = 0.0130) (Fig 2D), the same with tetanus toxoid ($P = 0.0049$); but for RH22030, the mean cord levels was higher for the placental malaria positive group ($P = 0.013$).

## Comparison of the risk of malaria for whole percentile versus 75th percentile of cord total IgG against selected *P.falciparum* specific antigens

The adjusted Hazard ratios for the 75th percentile of total IgG against *P. falciparum* specific antigens where hazards ratios were significantly high were generally higher than non-categorized/whole percentile; (Rh4.2; whole value = 1.08,75th percentile = 1.092, PfSEA; whole value = 1.15,75th percentile = 1.32, Etramp5Ag1; whole value = 1.06,75th percentile = 1.21, AMA1; whole value = 1.049, 75th percentile = 1.25, GLURP181; whole value = 1.02, 75th percentile = 1.83, EBA175; whole value = 1.043, 75th percentile = 1.35) (Table 3). Similar difference was noted for IgG levels against other *P. falciparum* antigens where hazard ratios were not significantly high (higher for 75th percentile compared to whole value). The risk of malaria in a child during first birth year increases with increase in cord levels of IgG (Fig 2).

## 25th, 50th and 75th percentile plots of total cord serum IgG against *P. falciparum* specific antigens used in the Cox regression model

Percentile plots of log transformed cord serum total IgG levels against Rh4.2, PfSEA, AMA1, GLURP, Etramp5Ag1 and EBA175 was done in excel. Values of 75th percentile were highest, followed by 50th and least was 25th percentile across all antigens (Fig 3). Total IgG levels against GLURP was highest compared to other antigens across all percentiles (25th = 6.76, 50th = 8.69, 75th = 9.31) and least for total IgG levels against PfSEA across all percentiles (25th = 2.56, 50th = 2.94, 75th = 3.43) (Fig 3)

## Risk of malaria among individual with different levels of expression of total IgG against selected *P.falciparum* specific antigens adjusted for non-IgG covariates on risk of malaria infections in the first birth year

Increased levels (75th percentile) of cord blood total IgG against Rh4.2 antigen and maternal poverty combined significantly increases risk of malaria infections in the first birth year of a child (AHR: Rh4.2 = 1.092, poorest = 0.000) (Table 4A). Increased expression of cord blood total IgG against PfSEA, gestational malaria infections and poverty combined increased risk of

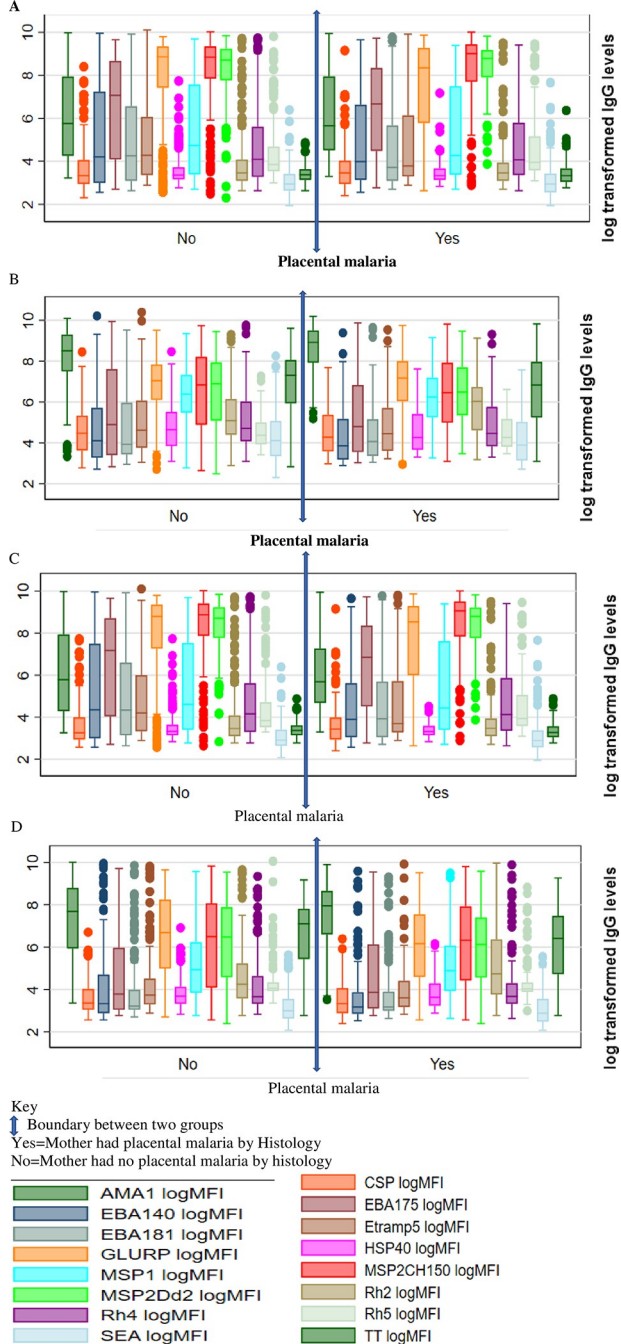

Key
Boundary between two groups
Yes=Mother had placental malaria by Histology
No=Mother had no placental malaria by histology

| | | |
|---|---|---|
| AMA1 logMFI | | CSP logMFI |
| EBA140 logMFI | | EBA175 logMFI |
| EBA181 logMFI | | Etramp5 logMFI |
| GLURP logMFI | | HSP40 logMFI |
| MSP1 logMFI | | MSP2CH150 logMFI |
| MSP2Dd2 logMFI | | Rh2 logMFI |
| Rh4 logMFI | | Rh5 logMFI |
| SEA logMFI | | TT logMFI |

**Fig 2. Effects of placental malaria on IgG sub-class cord expression.** (A) effects of placental malaria on IgG1 cord expression against *P.falciparum* specific antigens (B) effects of placental malaria on IgG2 cord expression against *P. falciparum* specific antigens; (C) effects of placental malaria on IgG3 cord expression against *P.falciparum* specific antigens; (D) effects of placental malaria on IgG4 transfer against malaria specific antigens- antibody levels measured in log MFI.

malaria infections in the early months of a child's existence (AHRs: PfSEA = 132, malaria was detected = 1.29 and maternal wealth index as poorest = 1.78) (Table 4B). When combined, high levels of cord blood total IgG against Etramp5Ag1 and maternal poverty significantly increases risk of malaria infections in the first birth year (AHRs: Etramp5Ag1 = 1.21, mother's

**Table 3. Hazard ratios of non-categorized percentile compared to 75th percentile of cord serum total IgG levels against selected *P.falciparum* antigen.**

| *P. falciparum* specific antigens | Hazard ratio for non-categorized percentile | 95% CI | Hazard ratio for 75th percentile | 95% CI |
|---|---|---|---|---|
| Rh 4.2 | 1.08 | 0.99, 1.11 | 1.09 | 1.02, 1.17 |
| PfSEA | 1.15 | 1.00, 1.33 | 1.32 | 1.00, 1.74 |
| Etramp5Ag1 | 1.06 | 0.99, 1.12 | 1.21 | 0.97, 1.52 |
| AMA1 | 1.05 | 0.99, 1.11 | 1.25 | 0.98, 1.60 |
| EBA175 | 1.04 | 0.98, 1.03 | 1.35 | 1.03, 1.78 |
| GLURP181 | 1.20 | 0.96, 1.08 | 1.83 | 1.15, 2.93 |

wealth index categorized as poorest = 1.78) (Table 4C). Similar findings were reported at increased expression of total levels against AMA1 adjusted for poverty. Risk of malaria infections significantly increased with increase in poverty and increase in IgG levels against AMA1 (AHRs: AMA1 = 1.25, mother's wealth index categorized as poorest = 1.87) (Table 4D). At increased level, serum cord total IgG levels against EBA175 combined with high maternal poverty and maternal malaria infections during pregnancy significantly increases risk of malaria infections during early days of a newborn existence (AHRs: EBA175 = 1.35, mothers' wealth index categorized as poorest = 1.86, mothers with malaria episodes during pregnancy = 1.30) (Table 4E). At increased levels, serum total IgG against GLURP combined with maternal poverty and effects of malaria infections during pregnancy significantly increases risk of clinical malaria disease in the first birth year of a child (AHRs: total IgG against GLURP = 1.83, malaria infection during pregnancy = 1.30, mothers' wealth index categorized as poorest = 1.79) (Table 4F)

## Discussion

We found no significant differences in cord IgG sub-type levels in children whose mothers were on SP versus those on DP for most antigens. Significant differences were only noted for tetanus toxoid in IgG1 levels against all the erythrocyte binding antigens (EBA140, EBA175,

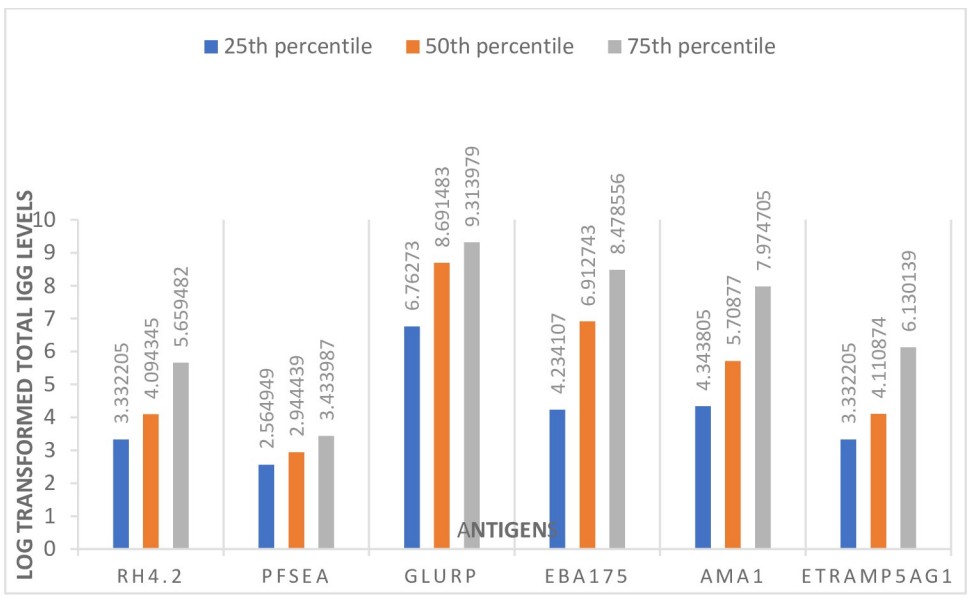

**Fig 3. Percentile plot of cord blood total IgG levels against *P.falciparum* specific antigens.**

**Table 4. Cox regression model of effects of increased levels of expression of total IgG against Rh4.2 adjusted for selected non-IgG covariates on risk of malaria infections in the first birth year.** A. B. Cox regression model of effects of increased levels of expression of total IgG against PfSEA adjusted for selected non-IgG covariates on risk of malaria infections in the first birth year. C. Cox regression model of effects of increased levels of expression of total IgG against Etramp5Ag1 adjusted for non-IgG covariates on risk of malaria infections in the first birth year. D. Cox regression model of effects of increased levels of expression of total IgG against AMA, adjusted for maternal wealth category index on risk of malaria infections in the first birth year. E. Cox regression models of effects of increased levels of expression of total IgG against EBA175, adjusted for non-IgG covariates on risk of malaria infections in the first birth year. F. Cox regression model of effects of increased levels of expression of total IgG against GLURP181 adjusted for non-IgG covariates on risk of malaria infections in the first birth year.

**A)**

| Variable | Categories | Un-Adjusted HR | 95% CI | Adjusted HR | P-value | 95% CI |
|---|---|---|---|---|---|---|
| logMFIRh4 | logMFIRh4 | 1.08 | 1.01, 1.15 | 1.09 | 0.01 | 1.02, 1.17 |
| Birthwgtcat | Normal | 1.00 | | 1.00 | | |
| | Low | 1.18 | 0.81, 1.72 | 1.43 | 0.14 | 0.90, 2.25 |
| Matprev | No malaria | 1.00 | | 1.0 | | |
| | Malaria detected | 1.40 | 1.11, 1.77 | 1.26 | 0.12 | 0.94, 1.70 |
| Gravidcat | 1–3 | 1.00 | | 1.0 | | |
| | 4+ | 0.94 | 0.73, 1.19 | 0.98 | 0.78 | 0.84, 1.14 |
| Wealthcat | Least poor | 1.0 | | | | |
| | Middle | 1.35 | 1.05, 1.75 | 1.24 | 0.18 | 0.90, 1.69 |
| | Poorest | 1.64 | 1.29, 2.09 | 1.86 | 0.00 | 1.36, 2.56 |

**B)**

| Variable | Categories | Un-adjusted HR | 95% CI | Adjusted HR | P-value | 95% CI |
|---|---|---|---|---|---|---|
| logMFISEAcat | logMFISEAcat | 1.20 | 1.03, 1.38 | 1.32 | 0.05 | 1.00, 1.74 |
| birthwgtcat | Normal | 1.00 | | 1.00 | | |
| | Low | 1.18 | 0.81, 1.73 | 1.41 | 0.14 | 0.89, 2.23 |
| matprev | No Malaria | 1.00 | | 1.00 | | |
| | Malaria detected | 1.40 | 1.11, 1.77 | 1.29 | 0.09 | 0.96, 1.73 |
| Gravidcat | 1–3 | 1.00 | | 1.00 | | |
| | +4 | 0.94 | 0.73, 1.19 | 0.98 | 0.90 | 0.722, 1.34 |
| Wealthcat | Least poor | 1.00 | | 1.00 | | |
| | Middle | 1.35 | 1.05, 1.75 | 1.26 | 0.16 | 0.92 1.72 |
| | Poorest | 1.64 | 1.29, 2.09 | 1.83 | 0.00 | 1.34 2.51 |

**C)**

| Variable | Categories | Un-adjusted HR | 95% CI | Adjusted HR | P-value | 95% CI |
|---|---|---|---|---|---|---|
| MFIEtramp5cat | MFIEtramp5cat | 1.22 | 0.99, 1.51 | 1.21 | 0.09 | 0.97, 1.52 |
| Birthwgtcat | Normal | 1.00 | | 1.00 | | |
| | Low | 1.18 | 0.81, 1.73 | 1.46 | 0.11 | 0.92, 2.31 |
| matprev | No malaria | 1.00 | | 1.00 | | |
| | Malaria detected | 1.40 | 1.11, 1.77 | 1.27 | 0.11 | 0.94, 1.72 |
| Gravidcat | 1–3 | 1.0 | | | | |
| | +4 | 0.94 | 0.73, 1.19 | 0.97 | 0.82 | 0.71, 1.32 |
| Wealthcat | Least poor | 1.00 | | 1.00 | | |
| | Middle | 1.35 | 1.05, 1.75 | 1.19 | 0.28 | 0.87, 1.62 |
| | Poorest | 1.64 | 1.29, 2.09 | 1.78 | 0.00 | 1.30, 2.43 |

**D)**

| Variable | Category | Un-adjusted HR | 95% CI | Adjusted HR | P-value | 95% CI |
|---|---|---|---|---|---|---|
| logMFIAMA1cat | logMFIAMA1cat | 1.24 | 0.96 1.59 | 1.25 | 0.07 | 0.98, 1.60 |
| Wealthcat | Least poor | 1.00 | | 1.00 | | |
| | Middle | 1.35 | 1.05 1.75 | 1.24 | 0.16 | 0.92, 1.70 |
| | Poorest | 1.64 | 1.29 2.09 | 1.87 | 0.00 | 1.38, 2.54 |

*(Continued)*

**Table 4.** (Continued)

**E)**

| Variable | Category | Un-adjusted HR | 95% CI | Adjusted HR | P-value | 95% CI |
|---|---|---|---|---|---|---|
| logMFIEBA175cat | logMFIEBA175cat | 1.35 | 1.03, 1.78 | 1.35 | 0.03 | 1.03, 1.78 |
| Wealthcat | Least poor | 1.00 | | 1.00 | | |
| | Middle | 1.35 | 1.05, 1.75 | 1.26 | 0.14 | 0.92, 1.71 |
| | Poorest | 1.64 | 1.29, 2.09 | 1.86 | 0.00 | 1.37, 2.52 |
| Matprev | No Malaria | 1.0 | | 1.0 | | |
| | Malaria detected | 1.40 | 1.11, 1.77 | 1.30 | 0.07 | 0.97, 1.74 |

**F)**

| Variable | Category | Un-adjusted HR | 95% CI | Adjusted HR | P-value | 95% CI |
|---|---|---|---|---|---|---|
| logMFIGLURP181cat | logMFIGLURP181cat | 1.81 | 1.13, 2.88 | 1.83 | 0.01 | 1.15, 2.93 |
| matprev | No Malaria | 1.00 | | 1.0 | | |
| | Malaria detected | 1.40 | 1.11, 1.77 | 1.30 | 0.08 | 0.96, 1.73 |
| Wealthcat | Least poor | 1.00 | | | | |
| | Middle | 1.35 | 1.05, 1.75 | 1.20 | 0.25 | 0.88, 1.64 |
| | Poorest | 1.64 | 1.29,2.09 | 1.79 | 0.00 | 1.31, 2.4 |

Key

Wealthcat = categorization of mothers' economic status

Matprev = categorization of gestational malaria prevalence

logMFIGLURP181cat = 75[th] percentile of logarithmic transformed levels of total IgG against GLURP181 antigen

logMFIEBA175cat = 75[th] percentile of logarithmic transformed levels of total IgG against EBA175 antigen

logMFIAMA1cat = 75[th] percentile of logarithmic transformed levels of total IgG against AMA1 antigen

MFIEtramp5cat = 75[th] percentile of logarithmic transformed levels of total IgG against Etramp5Ag1 antigen

Birthwgtcat = categorization of neonate's birth weight in grams

Gravidcat = categorization of gravidity status of the mother

logMFISEAcat = 75[th] percentile of logarithmic transformed levels of total IgG against PfSEA antigen

logMFIRh4 = 75[th] percentile of logarithmic transformed levels of total IgG against Rh4.2 antigen

EBA181) and IgG4 against merozoite surface proteins (MSPCH150, MSP2Dd2). In all cases where significant differences were observed, mean cord levels for SP group are higher than the DP group. While most of the previous studies haven't directly compared effects of DP and SP on cord expression of IgG subtypes against *P. falciparum* specific antigens, one study did compare total IgG cord levels for expectant mothers on prophylaxis and those who never used. They found a significant difference in the mean total IgG levels between the two groups with expectant mothers on prophylaxis expressing less total IgG against selected antigens compared to those mothers who were enrolled on none [17]. There are no documented effects of either anti-malaria prophylaxis on the biochemical properties such as affinity, avidity and titers of the antibodies, so either of them can be used to prevent acquisition of malaria during pregnancy with little or minimal effect on malaria specific antibody titer [20]. A study by Ariera in Kenya did not also indicate any direct effects of either SP or DP on either transfer or expression of total IgG and subclasses from placenta to cord blood [21]. This is probably due to the fact that neither of the two drugs compete for the same binding site on the FcRn of the syncytiotrophoblast that IgG and its sub-classes usually bind to [22]. The Ugandan Ministry of Health and other malaria endemic regions can benchmark on these findings to allow other anti-malaria prophylaxes to be utilized in the clinical setting as opposed to the current monopoly of SP. Currently in Uganda, the only recommended and approved anti-malaria prophylaxis

is SP (common brand is called Fansidar). Further studies are needed to explore if the difference between SP and DP for IgG4 against signature *P. falciparum* antigens can significantly affect child's progression to severe disease.

We found no association between placental malaria and the cord levels of IgG sub-class levels in malaria holoendemic area of Busia, Eastern Uganda. Even where differences exist, it was never skewed to either mothers who were negative for placental malaria or those positive but varied from one antigen to another. Studies done in other parts of Africa had different findings; some found strong association between placental malaria and reduced transfer/expression of maternal IgG from placental to cord blood. One study of interest concluded there was no significant difference in the expressed levels of antibodies against signature *P. falciparum* specific antigens between children born to mothers with placental malaria and those born to mothers without [23]. In other studies, the transfer and expression of total IgG, from placenta to cord blood was found to be increased in mothers who had placental malaria [24]. Most studies searched compared effects of at least two co-morbidities such as HIV and placental malaria; this made it difficult to attribute observed decrease in transfer or expression to placental malaria [14, 25, 26]. A previous study did not also find any association between placental malaria and cord antibody levels [27]. Repeated episodes of malaria, placental malaria infection is proven to dampen the immune system leading to immunological tolerance [28]. This can partly explain lack of association between placental malaria and cord levels of IgG sub-classes against *P. falciparum* specific antigens [29, 30].

The multivariate cox regression analysis showed children with increased levels of cord blood total IgG against PfSEA, Rh 4.2, AMA1, EBA175, Etramp5Ag1and GLURP antigens have increased risk of malaria infection in the first birth year. The Schizont Egress 1 antigen facilitates merozoites egress; the protein has also been associated with kinetochore function during schizont development [31]. All rhoptry proteins (including Rh4.2) have been shown to be the dominant proteins transferred to the surface of normal erythrocytes during delayed or aborted re-invasion. Antibodies to this protein are known to mediate protection in immunized monkeys and to inhibit parasite growth *invitro* [32]. AMA1 is abundantly expressed in sporozoites and has been associated with parasite invasion of the liver cells (hepatocytes) [33]. *Etramps* (including Etramp5Ag1) are important mediators of plasmodium- host cell interactions while playing significant roles in parasite pathogenesis [34]. Expression of antibodies against these key *P. falciparum* antigens without protective could potentially increase the risk of malaria infections, with high likelihood of progression to severe disease. High incidence of gestational malaria also increases risk of malaria infection in a child during first months of existence. This could be because of exposure to similar source of infections like the mother. Early in life, the lifestyle of the baby is tied to that of mother and have similar exposures to malaria prevention strategies for example, sleeping under the same ITNs, sleeping in the same IRS house among others. Findings by this study agrees with those of other scholars and researchers [25, 35].

This study found that children born to poor mothers had highest risk of malaria infections during the first birth year. Mothers with inadequate income cannot afford quality food and nutrition supplement required for a baby's growth, cannot promptly diagnose and treat infections such as malaria, parasitic and bacterial infections and others that can affect a baby's health [36]. Such mothers cannot afford other mosquito vector control measures such as repellants and sprays. Some studies found fever (a proxy measure of malaria) to affects children from very poor and least poor families in near equal proportion [37]. One prospective cohort study in rural areas of Tanzania found mortality from malaria among children born to very poor families were higher than those among children born to least poor families [38].

### Limitation of the study

Most children were breast feeding, so we could not account for IgG taken through breast milk. The roles of other cofounding such as IgGA in breast milk, lactoferritin in breast milk, and fetal Hemoglobin in clearing parasitemia were not adjusted for during this study. We had a challenge with deciding on the appropriate concentration of secondary antibody to be used in the Luminex assay because the original dilution factor in both the protocol and manufacturer's instruction were not yielding very good results.

## Conclusion

Malaria prophylaxis in pregnant mothers using either DP or SP does not affect expression of antibodies against *P. falciparum* specific antigens in the cord blood. Poverty and malaria infections during pregnancy are key risk factors of malaria infections in the early days of growth of children born in areas of high malaria transmission such as Uganda. Antibodies against *P. falciparum* does not protect against parasitemia and malaria infections among children born in the disease endemic areas.

## Supporting information

**S1 Appendix.**
(PDF)

**S2 Appendix.**
(PDF)

**S1 File.**
(DO)

**S2 File.**
(DTA)

**S3 File.**
(DTA)

**S4 File.**
(DTA)

**S5 File.**
(DTA)

## Acknowledgments

First, I want to thank Dr. Isaac Sewanyana for the great mentorship he offered to me during my master's degree study regarding malaria immunology, Dr Moses Ocan, thank you for teaching me the very basics of concept writing, literature review and scientific writing in general. We sincerely appreciate Makerere University Malaria Research training program under the leadership of Prof. R. Kamya and your entire team (Dr. Joaniter Nankabirwa and Dr. Emmanuel Arinaitwe) for sponsoring this study. The team in a special way wants to thank Associate Professor Prasana Jaganathan and August Nicholas Zehner of Stanford University (USA) for providing coupled beads and offering and intensive hands-on technical training and expertise for the Ugandan team. Madam Rhoda Namubiru, thank you for providing finances in time always. Great votes of thanks go to the Ugandan Ministry of Health under the leadership of the Hon. Minister Dr. Jane Ruth Aceng for providing part of tuition during the

earlier years of my MSc degree program. Mapronano African Centre of Excellence under the leadership of Prof. Kirabira and Brian Mujuni also offered a much-needed financial boost in the middle of the project, we honestly thank you. Central Public Health Laboratories under the leadership of Dr.Suzan Nabadda and Global Health Laboratories under the leadership of Mr, Waswa thank you for offering your Laboratories to conduct this study and run our Luminex assays.The team also acknowledges the contribution of the entire Makerere University Master of Science-Immunology and Microbiology class of 2017, the entire staffs of the department under the great leadership of Dr.David Patrick Kateete, Dr Obondo James Sand and Dr. Bwanga Freddie who was the course coordinator. Finally, let me acknowledge my wife (stella), kids (Daisy, Arthur, and Pauline), my mother (Margaret) and father (Allan)

## Author Contributions

**Conceptualization:** Erick Jacob Okek, Isaac Ssewanyana.

**Data curation:** Erick Jacob Okek.

**Formal analysis:** Erick Jacob Okek, Joaniter Nankabirwa, Isaac Ssewanyana.

**Funding acquisition:** Moses Robert Kamya.

**Investigation:** Erick Jacob Okek, Anthony Kiyimba, Emmanuel Arinaitwe.

**Methodology:** Erick Jacob Okek.

**Supervision:** Moses Ocan, Sande James Obondo, Isaac Ssewanyana.

**Validation:** Isaac Ssewanyana.

**Writing – original draft:** Erick Jacob Okek.

**Writing – review & editing:** Sande James Obondo.

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
