## [Decision Letter · Decision Letter 0]

27 Jan 2022

PONE-D-21-38887Maternal transfer of IgG sub-types and its association to Immunity against Plasmodium falciparum in a Ugandan birth cohort (Busia-Eastern Uganda)PLOS ONE

Dear Dr. Okek,

Thank you for submitting your manuscript to PLOS ONE. After careful consideration, we feel that it has merit but does not fully meet PLOS ONE’s publication criteria as it currently stands. Therefore, we invite you to submit a revised version of the manuscript that addresses all the points raised below during the review process.

We look forward to receiving your revised manuscript.

Kind regards,

Ray Borrow, Ph.D., FRCPath

Academic Editor

PLOS ONE

Journal Requirements:

6. We note that Figure 1 in your submission contain [map/satellite] images which may be copyrighted. All PLOS content is published under the Creative Commons Attribution License (CC BY 4.0), which means that the manuscript, images, and Supporting Information files will be freely available online, and any third party is permitted to access, download, copy, distribute, and use these materials in any way, even commercially, with proper attribution. For these reasons, we cannot publish previously copyrighted maps or satellite images created using proprietary data, such as Google software (Google Maps, Street View, and Earth). For more information, see our copyright guidelines: http://journals.plos.org/plosone/s/licenses-and-copyright.

8. Please amend the manuscript submission data (via Edit Submission) to include author Kiyimba Anthony, Arinaitwe Emmanuel, Nankabirwa Joaniter, and Kamya.R.Moses.

9. Please include your full ethics statement in the ‘Methods’ section of your manuscript file. In your statement, please include the full name of the IRB or ethics committee who approved or waived your study, as well as whether or not you obtained informed written or verbal consent. If consent was waived for your study, please include this information in your statement as well. 

10. Please ensure that you refer to Figure xxxxx in your text as, if accepted, production will need this reference to link the reader to the figure.

11. Please include captions for your Supporting Information files at the end of your manuscript, and update any in-text citations to match accordingly. Please see our Supporting Information guidelines for more information: http://journals.plos.org/plosone/s/supporting-information. 

Reviewers' comments:

Reviewer's Responses to Questions

**Comments to the Author**

1. Is the manuscript technically sound, and do the data support the conclusions?

Reviewer #1: Yes

Reviewer #2: Yes

Reviewer #3: Partly

2. Has the statistical analysis been performed appropriately and rigorously? 

Reviewer #1: No

Reviewer #2: Yes

Reviewer #3: Yes

3. Have the authors made all data underlying the findings in their manuscript fully available?

Reviewer #1: Yes

Reviewer #2: Yes

Reviewer #3: No

4. Is the manuscript presented in an intelligible fashion and written in standard English?

Reviewer #1: Yes

Reviewer #2: Yes

Reviewer #3: No

5. Review Comments to the Author

Reviewer #1: This study evaluated effects of Intermittent Prophylactic Treatment in Pregnancy and placental malaria status on malaria specific IgG levels in the fetus and the association between IgG levels and occurrence of malaria in the first year of the infants. This is an interesting and important research topic. I have some questions about the data analysis.

The authors mentioned that “Increasing antibody titer against P.falciparum may be more of a

marker of exposure to previous malaria infections”. What does “previous malaria” mean here? Mothers got malaria during pregnancy? Doesn’t placenta malaria capture that information?

For the survival analysis what if the placenta malaria status was used as the grouping variable instead of IgG levels?

Does “ANOVA in Box plot” means the data are plotted in box plot and comparisons were done using ANOVA? ANOVA compares means instead of medians. Because there are only two groups in the comparison, ANOVA is the same as t-test. Also many of the log-IgG level seem to have a skewed distribution, a non-parametric test may be needed. Due to the relatively large number of comparisons (4 IgG subtypes and 15 antigens) P-values need to be adjusted for multiple testing.

IgG levels may be affected by many factors, e.g., SP vs DP, placenta malaria status, clinical characteristics of the mothers. A multivariable regression model will be helpful to identify factors that may influence IgG levels.

In the result section about survival curves, hazard ratio (HR) were given at the end of the paragraph but which IgG was this? Does the star in Figure 3 means p<0.05 between the two groups? If so, stars were seen in most of the subplots but only a few HRs were mentioned in the text. In addition, it will be interesting to fit a Cox regression model to assess the association while adjusting other covariates in the model. The low concentration group had a lower risk comparing to the higher concentration group. Shouldn’t the HR be <1 rather than >1. Please double check the text/numbers.

Are data on the mother’s social economic status (SES) available? If so, it would be interesting to include the SES variables in the regression models that predicting IgG levels and the Cox model to see if they play a role.

Reviewer #2: The manuscript is very well written, and the results are very interesting. However, the authors could make some corrections.

Manuscript Title:

Authors can revise the title and mention the effect of treatment with SP and DP, instead of talking only about P. falciparum infection

Abstract

- Choose between antibody and anti-body

- Results: Specify the difference (high or low) in IgG4 between children from mothers on SP vs DP, same thing for EBA

- Conclusion: " ncreasing antibody titer against P.falciparum " in whom? Please specify

Introduction

- Antibody dependent cellular cytotoxicity (ADCC) - not antigen dependent cellular cytotoxicity

- Put space before the references or not. Please harmonize

- High expression of IgG4 has been associated with repeated exposure to specific…of IgG4: Please put reference

Materials and methods

- Participants and data collection: Give enough details

- Describe preparation of buffer A, then B

- MagPix multiplex bead array assay: If available, please provide the IgG positivity threshold and specify how this threshold was determined

- The sample collection times for carrying out the survival tests were not detailed.

Results

- Figure 1: Review the figure, with good resolution.

- Why does the representation of the graphs differ? Please harmonize

- Specify the number of participants per group (n = ….)

- Association of placental malaria and malaria specific IgG antibodies:

Is there not an effect of both treatment (IPTp) / placental infection with P.f on the level of IgG?

Did all women receive IPTp treatment?

The authors can compare the level of IgG in children from mothers on SP (P.f neg vs P.f pos), and in children from

mothers on PD (P.f neg vs P.f pos).

- Association of malaria specific IgG sub-types and incidence of malaria in the study population

Authors can change “probability of survival” to “probability of fewer occurrence of malaria”

Discussion :

The authors can provide explanations for the differences observed only with a few antigens. Do these antigens have any particularities? The results obtained must be well discussed.

Reviewer #3: Erick and colleagues presents a report investigating levels of IgG in pregnant women cord blood receiving IPTp (or not) and in their infants after birth. They further attempt to investigate if IgG levels in cord blood has a protective effect against malaria. Similar studies have been performed previously, but this study includes a broad panel of 15 different Pf IgG targets and does well to look at IgG subclasses.

Major comments

There are numerous grammatical errors throughout the text, figures, and tables. The authors need to have their manuscript reviewed by someone with English as their first language in order to provide a clearer presentation of their findings.

Minor comments

Abstract and throughout: make sure to specify that you are assaying only for IgG antibodies. For example, the phrase in the Abstract: "Cord concentrations of the erythrocyte binding antigens (EBA140, EBA175 and EBA181) were significantly different.." makes it sound like you are detection the actual Pf EBAs, and not the Ig against EBAs.

Throughout: the assay signal for IgG is never translated to a concentration (ug/mL for example), so the authors should not refer to "concentrations" of antibodies in their study. Could replace with a word like "levels"

Introduction:

- "Mandatory" for Ugandan women to be enrolled in IPTp?

- Can delete "and consumes most of this essential gas"

- Please modify "most importantly the role of IgG from plasma.." as Ig transfer is one of many protective factors and shouldn't be assigned as the most important

- Can delete "(total and sub-classes)" since this is redundant

- "These anti-malarial immunoglobulins have varied..."

- "Antibodies against merozoite surface proteins are the most expressed in cord blood". You can just say cord blood has high levels of anti-MSP antibodies.

- DP and SP need to be written out at first use.

Materials and Methods:

- Need to explain specifically how the blood samples were collected, stored short-term (what type of containers, coagulant, fractionated?), and temperature stored at long-term until the IgG assay. It appears plasma was the sample stored long-term.

- Provide company information for all lab chemicals and assay reagents

- Need to provide references for the antigens use in the panel, and more detail on coupling conditions.

- Need to provide software that was used for analysis, and how the Kaplan-Meir curves were created in the software

Results:

- Figure 1: why are all 4 panels not boxplots?

- Figure 1: Missing AMA1 for panel A

- Figure 1: is panel B for IgG2 or IgG3?

- Figure 3: need all x and y axes to be the same among panels and plots. Some x axes appear to be days, and some months. Y axis isn't "Probability of Survival" but "Probability of no malaria infection"

- Figure 3 legend: panel A is IgG1 correct?

Discussion:

- "so either of them can be used safely to prevent acquisition of malaria..." the safety of these drugs has nothing to do with IgG antibody levels in women taking them. Please revise.

- "We found no overall association between placental..."

- Please remove the sentence referencing hemozoin as it is incorrect and not relevant here.

- "cofounders such as IgGA"?

6. PLOS authors have the option to publish the peer review history of their article (what does this mean?). If published, this will include your full peer review and any attached files.

Reviewer #1: No

Reviewer #2: No

Reviewer #3: No

---

## [Author Response · Author response to Decision Letter 0]

26 Mar 2022

Response to gaps identified and guidance offered following submission of Manuscript to PLOS ONE publication. 

The authors of the manuscript would wish to sincerely appreciate your honest and candid feedback following the article submitted to PLOS ONE journal for reviews, edits and eventual publications. We have sat down as a team and made adjustments and formatted the manuscript to conform to your guidance and editorial reviews; details of gaps identified, and corresponding responses are detailed below

Comment #1, Editor

Response: We take note of your guidance; the link shared has been so helpful in adjusting the manuscript to conform to PLOS ONE publication requirement. The revised manuscript re-submitted now meets standards for publication in PLOS ONE Journal. The revised Manuscript has been adjusted to conform to PLOS ONE publication 

Comment#2, Editor:We note that the grant information you provided in the ‘Funding Information’ and ‘Financial Disclosure’ sections do not match

Response: . Thanks for the comment; information on the funding and financial disclosure section has been revised in the re-submitted manuscript 

Comment #3, Editor: We note that you have stated that you will provide repository information for your data at acceptance. Should your manuscript be accepted for publication, we will hold it until you provide the relevant accession numbers or DOIs necessary to access your data. If you wish to make changes to your Data Availability statement, please describe these changes in your cover letter and we will update your Data Availability statement to reflect the information you provide.

Response: Thanks for the observation, the data availability statement has been revised to indicate that, all the data for the study will be available on request from the author upon publication of the manuscript. This information has been included in the revised cover letter

Comment #4, Editor: PLOS requires an ORCID iD for the corresponding author in Editorial Manager on papers submitted after December 6th, 2016. Please ensure that you have an ORCID iD and that it is validated in Editorial Manager. To do this, go to ‘Update my Information’ (in the upper left-hand corner of the main menu), and click on the Fetch/Validate link next to the ORCID field. This will take you to the ORCID site and allow you to create a new iD or authenticate a pre-existing iD in Editorial Manager. Please see the following video for instructions on linking an ORCID iD to your Editorial Manager account: https://www.youtube.com/watch?v=_xcclfuvtxQ

Response: Thanks for the guidance, the ORCID ID for the corresponding author (: 0000-0002-2836-714 has been provided in the revised manuscript. 

Comment #5, Editor: Please include your full ethics statement in the ‘Methods’ section of your manuscript file. In your statement, please include the full name of the IRB or ethics committee who approved or waived your study, as well as whether you obtained informed written or verbal consent. If consent was waived for your study, please include this information in your statement as well. 

Response:. Our current study used samples that were collected from a previous study that was reviewed and cleared by School of Biomedical Sciences Research and Ethics Committee (SBS 114). The samples from primary study were collected after obtaining a written informed consent from study participants. This current study obtained ethical clearance from School of Biomedical sciences Research and Ethics Committee, College of Health Sciences Makerere University (SBS 1012). The archived samples used for this study, were de-identified prior to being accessed by the study team.

Comment #6, Editor: We note that Figure 1 in your submission contain [map/satellite] images which may be copyrighted. All PLOS content is published under the Creative Commons Attribution License (CC BY 4.0), which means that the manuscript, images, and Supporting Information files will be freely available online, and any third party is permitted to access, download, copy, distribute, and use these materials in any way, even commercially, with proper attribution. For these reasons, we cannot publish previously copyrighted maps or satellite images created using proprietary data, such as Google software (Google Maps, Street View, and Earth). For more information, see our copyright guidelines: http://journals.plos.org/plosone/s/licenses-and-copyright

Response: Thanks for the observation, however, fig1 was developed from the analysis of the data generated from our study. The figure 1 is not a map but was generated from the analysis of our study data. 

Comment #7, Editor: Please include captions for your Supporting Information files at the end of your manuscript, and update any in-text citations to match accordingly. Please see our Supporting Information guidelines for more information: http://journals.plos.org/plosone/s/supporting-information. 

Response: . Thank you for the observations; proposed corrections have been effected in the revised manuscript

Comment #8 Editor: Please amend the manuscript submission data (via Edit Submission) to include author Kiyimba Anthony, Arinaitwe Emmanuel, Nankabirwa Joaniter, and Kamya.R.Moses.

Response: Thanks for observation, the authors were omitted from the initial submission, but have been included in the revised manuscript.

Comment#9 Editor: Please ensure that you refer to Figure xxxxx in your text as, if accepted, production will need this reference to link the reader to the figure.

Response: This guidance is not very clear, there is no such figure as xxxxx in my text; However, we have referenced all the figures in the main text of the revised manuscript.

Comment #10, Editor: Please include captions for your Supporting Information files at the end of your manuscript, and update any in-text citations to match accordingly. Please see our Supporting Information guidelines for more information: http://journals.plos.org/plosone/s/supporting-information. 

Response: Thanks for the comments; this has been effected in the revised manuscript. 

Reviewer’s comments

Reviewer One:

Comment#1: Reviewer 1: The authors mentioned that “Increasing antibody titer against P.falciparum may be more of a

marker of exposure to previous malaria infections”. What does “previous malaria” mean here? Mothers got malaria during pregnancy. Doesn’t placenta malaria capture that information?

Response: Thank you for the comment, we acknowledged the lack of clarity in usage of the term “Previous malaria infections”; however, we intended to mean that the antibody titre levels observed amongst study participants were due to an exposure to Plasmodium falciparum parasites rather than being an indicator of protective immune response. The term “previous malaria” has been replaced by exposure to Plasmodium falciparum parasites. 

Comment#2; Reviewer 1: For the survival analysis what if the placenta malaria status was used as the grouping variable instead of IgG levels?

Response: In our study, we sought to assess how malaria infections in pregnancy affects malaria episodes in children born to these mothers. Therefore, measurements of IgG level was done as a marker of malaria infection in children. The IgG levels is thus the most appropriate grouping variable. If we use placental malaria as a grouping variable, it would be difficult to test our hypothesis that malaria specific IgG levels protects against both infection and severe disease in the first few months of life. 

Comment#3; Reviewer 1: Does “ANOVA in Box plot” means the data are plotted in box plot and comparisons were done using ANOVA? ANOVA compares means instead of medians. Because there are only two groups in the comparison, ANOVA is the same as t-test. Also, many of the log-IgG level seem to have a skewed distribution, a non-parametric test may be needed. Due to the relatively large number of comparisons (4 IgG subtypes and 15 antigens) P-values need to be adjusted for multiple testing.

Response: 

Thanks for the comment, we have now analyzed the data using, Man-Whitney u test (a non-parametric test) has been used to compare means of IgG sub-type levels across DP and SP in the revised manuscript. Independent variable was treatment arm while dependent variable was IgG sub-type levels against different P. falciparum antigens. 

Comment #4; Reviewer 1: In the result section about survival curves, hazard ratio (HR) was given at the end of the paragraph, but which IgG was this? Does the star in Figure 3 means p<0.05 between the two groups? If so, stars were seen in most of the subplots but only a few HRs were mentioned in the text. In addition, it will be interesting to fit a Cox regression model to assess the association while adjusting other covariates in the model. The low concentration group had a lower risk comparing to the higher concentration group. Shouldn’t the HR be <1 rather than >1. Please double check the text/numbers.

Response: Noted; it is true that hazard ratio stated is only for IgG1 because it is the IgG sub class that plays the most vital role in malaria immunology; reciprocal Hazard ratio is what has been presented explaining why values are more than one.

Comment #5; Reviewer 1: Are data on the mother’s social economic status (SES) available? If so, it would be interesting to include the SES variables in the regression models that predicting IgG levels and the Cox model to see if they play a role.

Response: Yes, Social Economic status has been captured; these includes wealth index category (poor, very poor, average, well off), Education status type of house construction, and use of long-lasting insecticide treated mosquito net. Previous studies that attempted to look at effects of social economic status of mothers on cord expression of IgG levels in newborn did not have significant findings or association between SES and cord IgG sub-class expression

Reviewer 2:

Comment #1; Reviewer 2: Authors can revise the title and mention the effect of treatment with SP and DP, instead of talking only about P. falciparum infection

Response: Thanks for the comment, our revised tittle now reads; “Effects of anti-malarial prophylaxes on maternal transfer of Immunoglobulin-G (IgG) and association to Immunity against Plasmodium falciparum infections among Children in a Ugandan birth Cohort”

Comment #2 Abstract; Reviewer 2: Choose between antibody and anti-body

Response: Authors have chosen to use “antibody” not “anti-body”; this has been corrected all throughout the revised manuscript

Comment #3 Abstract, Reviewer 2: Results: Specify the difference (high or low) in IgG4 between children from mothers on SP vs DP, same thing for EBA

Comment #4 Abstract,Reviewer 2: Conclusion: " increasing antibody titer against P.falciparum " in whom? Please specify

Response: Thanks for the comment. This has been corrected in the revised manuscript, ‘increasing antibody titer against P. falciparum in children’ 

Comment #5, Introduction; Reviewer 2: 

- Antibody dependent cellular cytotoxicity (ADCC) - not antigen dependent cellular cytotoxicity

- Put space before the references or not. Please harmonize

- High expression of IgG4 has been associated with repeated exposure to specific…of IgG4: Please put reference

Response: Statement has been changed from antigen dependent cellular cytotoxicity to antibody dependent cellular cytotoxicity as guided by the reviewer. 

Comment #6: Reviewer 2: Materials and methods

- Participants and data collection: Give enough details

- Describe preparation of buffer A, then B

- MagPix multiplex bead array assay: If available, please provide the IgG positivity threshold and specify how this threshold was determined

- The sample collection times for carrying out the survival tests were not detailed.

Response: Demographic and clinical details of study participants is captured in a previous study which was published https://doi.org/10.1016/S0140-6736(18)32224-4; it has been cited as well .Description of Preparation of buffer A was made first, that of buffer B then followed as guided by the reviewer. MagPix multiplexbead array assay: The plates were read on a Magpix platform (Luminex, USA), acquiring at least 50 beads/region/well. The results were expressed as median fluorescence intensity (MFI). The blank well MFI (background effects) was deducted from each well to determine the net MFI (IgG positive result). Sample collection for Kaplan Meir survival curve analysis

Primary study where samples for this study were retrieved from was collected during delivery (cord blood). Collection was done by the midwife, medical officer or obstetrician who conducted delivery or cesarian section. Average of 4mls of accompanying mother’s venous blood was collected as well in an EDTA tube (mother baby pair). Both cord and maternal blood were centrifuged to obtain serum, archived at appropriate temperature, and retrieved for this study. Archival was done in mother-baby pair arrangement, a random-access number generated by the computer was used to track samples in the biobank. Venous blood samples were collected from infants at different time points during first year of life as and when malaria related signs and symptoms such as fever, vomiting or chills appear. Blood samples obtained was prepared and examined for presence of P.falciparum parasites; balance archived and used for future study like this one. Obtaining samples at different time points allowed for estimation of antibody decay rates.

Preparation of buffer A

Buffer A was prepared as discussed previously, but briefly, 1L of phosphate buffer solution was measured and put in a conical flask after which 500µl of tween was dispensed into the PBS to form PBS-tween solution.5g of PVP powder was weighed using a weighing balance in aluminum foil and added to the above solution after which equal amount of PVA powder was weighed and added to the mixture; proper agitation and mixing was done using a vortexer. 5mls of BSA was pipetted and added to the solution; finally, 0.2g of Sodium azide was weighed under a biosafety hood in an aluminum foil, added to the solution, properly vortexed and mixed. The final solution labelled with day of preparation and names of those who prepared.

Preparation of buffer B

A method by (18) was used in the preparation of buffer B. Briefly, 1000mls of phosphate buffer solution (PBS) was transferred to a sterile glass tube. To this was added, 500ul of tween solution (from thermos scientific), 5g of polyvinyl alcohol (PVA), 5mls of bovine serum albumin solution (BSA), 0.2g sodium azide and 3mls of lyophilized E.coli solution. The mixture was incubated overnight at 4oC. 

 IgG positivity rate was determined by subtracting combined sum of background effects and antibody MFI of pooled samples of six Caucasians from UK with no history of exposure to malaria from antibody MFI of blood samples of children born to mothers in Malaria endemic areas.

Comment #7 : Reviewer 2: Results

- Figure 1: Review the figure, with good resolution.

- Why does the representation of the graphs differ? Please harmonize

- Specify the number of participants per group (n = ….)

- Association of placental malaria and malaria specific IgG antibodies:

Is there not an effect of both treatment (IPTp) / placental infection with P.f on the level of IgG?

Did all women receive IPTp treatment?

The authors can compare the level of IgG in children from mothers on SP (P.f neg vs P.f pos), and in children from

mothers on PD (P.f neg vs P.f pos).

- Association of malaria specific IgG sub-types and incidence of malaria in the study population

Authors can change “probability of survival” to “probability of fewer occurrence of malaria”

Response: Figure was reviewed as guided the corrections are found in our revised submission 

- Figure one in the revised manuscript new have better resolutions and more clear graphical representation

-All plots in figure 1 have all been standardized.

-A total of 622 samples from infants were included in this study. Of these, 271 IgG1 results were merged with clinical data base, 305 IgG2 results were merged with clinical data base, 260 IgG3 results were merged with clinical data base and 269 IgG IgG4 results were merged as well.

-All women received IPTq because it is a policy in Uganda for all pregnant. The approved IPTq by the Ugandan Ministry of Health is three doses of SP (common brand is Fansidar) divided across the three trimesters of pregnancy. 

-studying effects of both treatment (IPTq) and placental infection with P.f on levels of IgG would be informative but was not the focus of our current study, this can be considered for future studies.

-Yes, all women received IPTq treatment; design of the original study randomized expectant mothers into the two arms of the prophylaxes; DP and SP. DP is the trial drug for IPTq while SP is the approved drug for routine use.

-suggestion of comparing IgG level in children from mothers on SP versus DP according to malaria status is a good one, however this was not done in the current study. 

- 

Comment #8; Reviewer 2 Discussion :

The authors can provide explanations for the differences observed only with a few antigens. Do these antigens have any particularities? The results obtained must be well discussed.

Response: The study was unique because of the large range of antigen parameter; dropping others means diluting the study. Discussion section has been expounded to capture reviewer’s guidance.

Particularities available include stage of infection where the parasite produced it, purification methods and class in which it belongs.

Reviewer 3:

Comment 1: Reviewer 3: Major comments

There are numerous grammatical errors throughout the text, figures, and tables. The authors need to have their manuscript reviewed by someone with English as their first language in order to provide a clearer presentation of their findings.

Response: This is noted in good faith, keen attention has been put in the revised manuscript to try and minimize grammatical and typing errors. Presentations in terms of tables, graphs will be clearer submitted this time

Comment 2: Reviewer 3 Minor comments

Abstract and throughout: make sure to specify that you are assaying only for IgG antibodies. For example, the phrase in the Abstract: "Cord concentrations of the erythrocyte binding antigens (EBA140, EBA175 and EBA181) were significantly different." makes it sound like you are detection the actual Pf EBAs, and not the Ig against EBAs.

Response: Comment noted; and revised to include reviewer’s guidance; malaria specific

Comment 3: Reviewer 3: Throughout: the assay signal for IgG is never translated to a concentration (ug/mL for example), so the authors should not refer to "concentrations" of antibodies in their study. Could replace with a word like "levels"

Response: This is noted, every sentence that has “concentration” has been changed to “level” in the revised manuscript.

Comment 4: Reviewer 3: Introduction:

- "Mandatory" for Ugandan women to be enrolled in IPTp?

- Can delete "and consumes most of this essential gas"

- Please modify "most importantly the role of IgG from plasma.." as Ig transfer is one of many protective factors and shouldn't be assigned as the most important

- Can delete "(total and sub-classes)" since this is redundant

- "These anti-malarial immunoglobulins have varied..."

- "Antibodies against merozoite surface proteins are the most expressed in cord blood". You can just say cord blood has high levels of anti-MSP antibodies.

- DP and SP need to be written out at first use.

 Response: Yes, it a policy that every pregnant mother be given some IPTp at least in every trimester

-“most important” has been deleted as advised by the reviewer.

-“total and sub-classes” deleted as advised

-both reviewer’s input in the last two bullets have been captured in the revised manuscript submitted.

-DP and SP has been quoted right from the beginning of introduction as guided by the reviewer

Comments 5: Reviewer 3: Materials and Methods:

- Need to explain specifically how the blood samples were collected, stored short-term (what type of containers, coagulant, fractionated?), and temperature stored at long-term until the IgG assay. It appears plasma was the sample stored long-term.

- Provide company information for all lab chemicals and assay reagents

Response.

Primary study where samples for this study were retrieved from was collected during delivery (cord blood) (reference). Collection was done by the midwife, medical officer or obstetrician who conducted delivery or cesarian section. Average of 4mls of accompanying mother’s venous blood was collected as well in an EDTA tube (mother baby pair). Both cord and maternal blood were centrifuged to obtain serum, archived at appropriate temperature, and retrieved for this study. Archival was done in mother-baby pair arrangement, a random-access number generated by the computer was used to track samples in the biobank. Venous blood samples were collected from infants at different time points during first year of life as and when malaria related signs and symptoms such as fever, vomiting or chills appear. Blood samples obtained was prepared and examined for presence of P.falciparum parasites; balance archived and used for future study like this one. Obtaining samples at different time points allowed for estimation of antibody decay rates.

-All Luminex reagents were made by Luminex Corp, Austin Texas USA. 

Comment 7: Reviewer 3: Need to provide references for the antigens use in the panel, and more detail on coupling conditions.

- Need to provide software that was used for analysis, and how the Kaplan-Meir curves were created in the software

Response: 

Graph pad prism was used to draw Kaplan -Meir curves, while box plots were created in STATA (ver 15). To create curve, immunological data (IgG1, IgG2, IgG3 and IgG4) were first merged with clinical data separately. Data was sorted in decreasing levels per IgG sub-type. Data was arranged into 25th percentile (portion containing least values) and 75th percentile (containing largest values) per IgG sub-class. The two groups of infants with varied antibody concentration were compared for occurrence of malaria in the first birth year. Malaria episodes was captured in the clinical data base.

IgG sub-types 1- 4 against 15 P. falciparum blood specific antigens and Tetanus Toxoid (TT) was measured in plasma diluted in buffer B. 50µl of bead suspension was added to each well (1,000 beads/region/well) of the 96 plate (Bio-plex pro-flat bottom). The plate was washed, placed on a magnetic block for 2 minutes and the supernatant was then poured off. The beads were washed twice with PBS tween buffer, laid on the magnetic separator for 2 minutes and supernatant poured off.

Reference for the antigens

Comment 8: reviewer 3: Results:

- Figure 1: why are all 4 panels not boxplots?

- Figure 1: Missing AMA1 for panel A

- Figure 1: is panel B for IgG2 or IgG3?

- Figure 3: need all x and y axes to be the same among panels and plots. Some x axes appear to be days, and some months. Y axis isn't "Probability of Survival" but "Probability of no malaria infection"

- Figure 3 legend: panel A is IgG1 correct?

Response.

All 4 panels in figure one has been revised to box plot

Figure 1 now has AMA1, question of whether panel B is for IgG2 or IgG3 has now been addressed in the revised manuscript.

Yes, panel A in Fig 3 is IgG1.

Mix up of scale in fig 3 has been corrected in the revised manuscript

Comment 9: reviewer 3: Discussion:

- "so either of them can be used safely to prevent acquisition of malaria..." the safety of these drugs has nothing to do with IgG antibody levels in women taking them. Please revise.

- "We found no overall association between placental..."

- Please remove the sentence referencing hemozoin as it is incorrect and not relevant here.

- "cofounders such as IgGA"?

Response: Statement containing safety of the two drugs has been removed from paragraph 1 of the discussion. 

This has been replaced by the statement “We found no overall association between placental” 

Statement referring to hemozoin has been deleted as guided by the reviewer

---

## [Decision Letter · Decision Letter 1]

20 Apr 2022

PONE-D-21-38887R1Effects of anti-malarial prophylaxes on maternal transfer of Immunogloublin-G (IgG) and  association to Immunity against Plasmodium falciparum infections among Children in a Ugandan birth CohortPLOS ONE

Dear Dr. Okek,

Thank you for submitting your manuscript to PLOS ONE. After careful consideration, we feel that it has merit but does not fully meet PLOS ONE’s publication criteria as it currently stands. Therefore, we invite you to submit a revised version of the manuscript that addresses the points raised below during the review process.

We look forward to receiving your revised manuscript.

Kind regards,

Ray Borrow, Ph.D., FRCPath

Academic Editor

PLOS ONE

Journal Requirements:

Reviewers' comments:

Reviewer's Responses to Questions

**Comments to the Author**

1. If the authors have adequately addressed your comments raised in a previous round of review and you feel that this manuscript is now acceptable for publication, you may indicate that here to bypass the “Comments to the Author” section, enter your conflict of interest statement in the “Confidential to Editor” section, and submit your "Accept" recommendation.

Reviewer #1: (No Response)

Reviewer #2: All comments have been addressed

Reviewer #3: All comments have been addressed

2. Is the manuscript technically sound, and do the data support the conclusions?

Reviewer #1: (No Response)

Reviewer #2: Yes

Reviewer #3: Yes

3. Has the statistical analysis been performed appropriately and rigorously? 

Reviewer #1: No

Reviewer #2: Yes

Reviewer #3: Yes

4. Have the authors made all data underlying the findings in their manuscript fully available?

Reviewer #1: Yes

Reviewer #2: Yes

Reviewer #3: Yes

5. Is the manuscript presented in an intelligible fashion and written in standard English?

Reviewer #1: Yes

Reviewer #2: Yes

Reviewer #3: Yes

6. Review Comments to the Author

Reviewer #1: Some of my previous questions/comments were not addressed in this revision which I will restate below. Due to the many factors that can influence malaria risk I still think a multivariate Cox regression is useful to understand the association.

A total of 16 Ig-G subtypes were used in the comparison and p-values need to be adjusted for multiple testing. I am not familiar with this particular research area, but would it be meaningful to combine all Ig-G subtypes into one variable in the analysis? This way multiple testing is not a concern and the main message will be easier to see. Right now there are too many comparisons which makes the result hard to interpret.

The conclusion from the survival model is that “antibody titer against P.falciparum may be more of a marker of exposure to previous malaria infections among infants from malaria endemic areas as opposed to their protective roles” which is interesting and that is why I recommended using placental malaria status as the grouping variable. If your conclusion is true I would expect to see higher malaria rates among those positive for placental malaria. The Cox model may also help to tease apart the effects of placental malaria and Ig-G levels.

Reviewer #2: The authors have taken into account the various comments and have done an excellent job in improving the manuscript.

Reviewer #3: The authors provide an improved version of the manuscript which is much easier to read and comprehend. A few minor issues need to be addressed:

- In Methods, the authors need to add company information for all of their chemicals, reagents, and other labware

- Concentrations of reagents such as secondary antibody and PE-conjugate used for the assays need to be included in Methods

- For Figures 1 and 2, include labels for the y axis on each panel. Perhaps on the x axis, can state "Placental Malaria", "No Placental Malaria", rather than "No", "Yes"

7. PLOS authors have the option to publish the peer review history of their article (what does this mean?). If published, this will include your full peer review and any attached files.

Reviewer #1: No

Reviewer #2: **Yes: **Odilon Paterne NOUATIN

Reviewer #3: No

---

## [Author Response · Author response to Decision Letter 1]

23 May 2022

Manuscript requirements

Please review your reference list to ensure that it is complete and correct. If you have cited papers that have been retracted, please include the rationale for doing so in the manuscript text, or remove these references and replace them with relevant current references. Any changes to the reference list should be mentioned in the rebuttal letter that accompanies your revised manuscript. If you need to cite a retracted article, indicate the article’s retracted status in the References list and also include a citation and full reference for the retraction notice

Response:

Thank you for the observation; We properly reviewed all citations and noted there was non which was incomplete or retracted from the journal. We noted that only citation number 23 and 25 were repeated and repeats were deleted

Reviewers’ comments

Reviewer #1, comment#1

Some of my previous questions/comments were not addressed in this revision which I will restate below. Due to the many factors that can influence malaria risk I still think a multivariate Cox regression is useful to understand the association.

Response: As guided by the reviewer, A multivariate cox regression using Breslow method for ties of the survival data was used to do this analysis. Failure variable was fever in a child in the first birth year, failure values were 3,4,5 which were episodes of fever while time to event was measured in days. The covariates included in the analysis are mother’s prophylactic arm (SP or DP), Insecticide treated mosquito use (used or not), placental malaria by histology (negative or positive) and total IgG levels against 16 malaria specific antigens

Reviewer #1, comment#2

A total of 16 Ig-G subtypes were used in the comparison and p-values need to be adjusted for multiple testing. I am not familiar with this research area, but would it be meaningful to combine all Ig-G subtypes into one variable in the analysis? This way multiple testing is not a concern, and the main message will be easier to see. Right now, there are too many comparisons which makes the result hard to interpret.

Response: Thank you for your keen observation; to add meaning to p-value during multivariate cox regression analysis, hazard ratio and z values were added to bring more meaning into the results.

Reviewer #1, comment#3

The conclusion from the survival model is that “antibody titer against P.falciparum may be more of a marker of exposure to previous malaria infections among infants from malaria endemic areas as opposed to their protective roles” which is interesting and that is why I recommended using placental malaria status as the grouping variable. If your conclusion is true, I would expect to see higher malaria rates among those positive for placental malaria. The Cox model may also help to tease apart the effects of placental malaria and Ig-G levels.

Response: we take note of this recommendation; In the cox regression analysis, Prescence of placental malaria by histology during delivery did not increase risk of developing fever in a child’s first birth year as there was no difference in number of fever episodes in children born to mothers with either placental malaria status (HR1. 1.3394,p=0.264 (table 3). We went further examined if difference in income levels affects birth weight of a child, time to first malaria incidence and number of malaria episodes. All three outcomes were aligned to mother’s economic status in a grouped column of graph pad prism. Two-way Analysis of Variance was used to compare difference in the mean values of the three groups across the three outcomes

Reviewer #3: The authors provide an improved version of the manuscript which is much easier to read and comprehend. A few minor issues need to be addressed:

Reviewer #3,comment #1

In Methods, the authors need to add company information for all their chemicals, reagents, and other labware

Response: We take note of your guidance; company names for almost reagents and chemicals used has been included in the revised manuscript

 Reviewer #3,comment #2

Concentrations of reagents such as secondary antibody and PE-conjugate used for the assays need to be included in Methods

Response: This has been factored in the revised manuscript in this statement “50µl of a secondary antibody specific for an IgG sub-type (IgG1, IgG2, IgG2, IgG3 and IgG4) with concentration of 1in1000 for IgG1,IgG2 and IgG4 while secondary antibody against IgG4 was diluted at 1in 2000 in dilution buffer, Using a multichannel pipette, 50µl of 1in 200 R-Phycoerythrin-conjugate AffinPure F (ab’) goat anti-human IgG

Reviewer #3,comment #2

For Figures 1 and 2, include labels for the y axis on each panel. Perhaps on the x axis, can state "Placental Malaria", "No Placental Malaria", rather than "No", "Yes"

Response: y axes have been labelled for all graphs as “log transformed IgG levels”; X-axis labelling has been adjusted in the revised manuscript.

---

## [Decision Letter · Decision Letter 2]

14 Jun 2022

PONE-D-21-38887R2Effects of anti-malarial prophylaxes on maternal transfer of Immunogloublin-G (IgG) and  association to Immunity against Plasmodium falciparum infections among Children in a Ugandan birth CohortPLOS ONE

Dear Dr. Okek,

Thank you for submitting your manuscript to PLOS ONE. After careful consideration, we feel that it has merit but does not fully meet PLOS ONE’s publication criteria as it currently stands. Therefore, we invite you to submit a revised version of the manuscript that addresses the points raised below during the review process.

We look forward to receiving your revised manuscript.

Kind regards,

Ray Borrow, Ph.D., FRCPath

Academic Editor

PLOS ONE

Reviewers' comments:

Reviewer's Responses to Questions

**Comments to the Author**

1. If the authors have adequately addressed your comments raised in a previous round of review and you feel that this manuscript is now acceptable for publication, you may indicate that here to bypass the “Comments to the Author” section, enter your conflict of interest statement in the “Confidential to Editor” section, and submit your "Accept" recommendation.

Reviewer #1: (No Response)

Reviewer #2: All comments have been addressed

Reviewer #3: (No Response)

2. Is the manuscript technically sound, and do the data support the conclusions?

Reviewer #1: (No Response)

Reviewer #2: Yes

Reviewer #3: Yes

3. Has the statistical analysis been performed appropriately and rigorously? 

Reviewer #1: (No Response)

Reviewer #2: Yes

Reviewer #3: Yes

4. Have the authors made all data underlying the findings in their manuscript fully available?

Reviewer #1: (No Response)

Reviewer #2: Yes

Reviewer #3: Yes

5. Is the manuscript presented in an intelligible fashion and written in standard English?

Reviewer #1: (No Response)

Reviewer #2: Yes

Reviewer #3: Yes

6. Review Comments to the Author

Reviewer #1: Cox model: why were only subjects with >=3 episodes of fevers considered as a failure instead of occurrence of malaria as in Figure 4? How were subjects with 1 and 2 episodes handled in the model? Family socioeconomic status and placental malaria status need to be adjusted for in the cox model. There are too many IgG variables in the model. Are different IgG levels correlated? If not, maybe better to fit each IgG in the model in turn and adjust the p-values for multiple testing. Table 3 needs to show variable description instead of variables names used in the coding. What is the overall P value? Is it likelihood ratio test of all the variables in the model? The likelihood ratio test should test for all the IgG related variables only.

It is not clear what is Figure 3A showing. Do the three tick marks on the x-axis in each subplot denote different outcome? What is the y-axis? Birth weight cannot be close to 0. Also it is not clear how the 2-way ANOVA was set up? If birth weight of a child, time to first malaria incidence and number of malaria episodes are the outcomes and mother’s economic status is a factor what is the other factor?

Result section on Kaplan-Meir survival curve, why was it called “reciprocal hazard ratio”? Higher level was associated with higher risk. Shouldn’t the value presented just be the hazard ratio for higher levels of IgG?

Reviewer #2: (No Response)

Reviewer #3: The authors still need to arrange some elements for easier interpretation. Table 3 should have defined rows and columns (like Tables 1 and 2) and the first 5 rows ("Time variable...." to "Overall P...") need to be deleted. In Figure 3, the two plots need clear labeling on the x and y-axes. For A, I assume the y-axis is birthweight, but what are the three categories on the x-axis? For B, I assume the x-axis is time to first malaria episode?

7. PLOS authors have the option to publish the peer review history of their article (what does this mean?). If published, this will include your full peer review and any attached files.

Reviewer #1: No

Reviewer #2: No

Reviewer #3: No

---

## [Author Response · Author response to Decision Letter 2]

21 Jul 2022

Review Comments to the Author

Reviewer #1: Cox model: why were only subjects with >=3 episodes of fevers considered as a failure instead of occurrence of malaria as in Figure 4? How were subjects with 1 and 2 episodes handled in the model?

Response: Thank you for the observation; we have dropped the number of fever episodes as a failure event and replaced it with malaria occurrence in a child throughout the observation period, failure event is now occurrence of malaria. This is adjusted for in the copy with tracked changes and the revised version. Failure values are 1,2,3,4,5 

 Family socioeconomic status and placental malaria status need to be adjusted for in the cox model. There are too many IgG variables in the model. Are different IgG levels correlated? If not, maybe better to fit each IgG in the model in turn and adjust the p-values for multiple testing. 

Response: Your comment is noted and addressed; all covariates (parasite prevalence by microscopy during pregnancy, malaria episodes during pregnancy, choice of IPT, placental malaria and total IgG response against malaria specific antigens) had their P-values adjusted for using Benjamin-Hoceberg correction for multiple testing in excel (Table 3)

Table 3 needs to show variable description instead of variables names used in the coding. What is the overall P value? Is it likelihood ratio test of all the variables in the model? The likelihood ratio test should test for all the IgG related variables only.

Response: Full names of variables have been captured below the table under the key. Variables full names are too long to fit in the variable column; readers will be referred to the key below table 3. The first 5 columns of table 3; including overall P value has been deleted as advised by reviewer 3; this adresses your concerns as well.

It is not clear what is Figure 3A showing. Do the three tick marks on the x-axis in each subplot denote different outcome? What is the y-axis? Birth weight cannot be close to 0. Also, it is not clear how the 2-way ANOVA was set up? If birth weight of a child, time to first malaria incidence and number of malaria episodes are the outcomes and mother’s economic status is a factor what is the other factor?

Response: Comments well noted; time to 1st malaria incidence and number of malaria incidence have been dropped off the variable list for the analysis; only birth weight has been maintained. Two way ANOVA in a grouped column graph was also dropped off as the analysis method; only mean data plot for birth weight versus economic status was maintained in the analysis (fig 3).

Result section on Kaplan-Meir survival curve, why was it called “reciprocal hazard ratio”? Higher level was associated with higher risk. Shouldn’t the value presented just be the hazard ratio for higher levels of IgG?

Thank you for the keen observation and guidance; It was indeed in error to refer to it as reciprocal Hazard ratio; it has been corrected to read Hazard ratio as it is the right statistical language.

Reviewer #2: (No Response)

Reviewer #3: The authors still need to arrange some elements for easier interpretation. Table 3 should have defined rows and columns (like Tables 1 and 2) and the first 5 rows ("Time variable...." to "Overall P...") need to be deleted. In Figure 3, the two plots need clear labeling on the x and y-axes. For A, I assume the y-axis is birthweight, but what are the three categories on the x-axis? For B, I assume the x-axis is time to first malaria episode?

Table 3 titled “multivariate cox regression of malaria occurrence in the first birth year of children associated with selected maternal characteristics/exposure variables” has clearly and well-defined titles within each column. For titles of rows, short forms, and acronyms of the full names have been used. Full names are captured under the key below the table. First five rows of table 3 deleted as guided. Figure 3A was removed as guided by the 1st reviewer, For Figure 3B, the two variables of time to first malaria and number of malaria episodes were dropped in the data mean set analysis; only birth weight was sustained as variable of comparison with income status. Figure 3 is now clearer and straight forward

---

## [Decision Letter · Decision Letter 3]

10 Aug 2022

PONE-D-21-38887R3Effects of anti-malarial prophylaxes on maternal transfer of Immunogloublin-G (IgG) and  association to Immunity against Plasmodium falciparum infections among Children in a Ugandan birth CohortPLOS ONE

Dear Dr. Okek,

Thank you for submitting your manuscript to PLOS ONE. After careful consideration, we feel that it has merit but does not fully meet PLOS ONE’s publication criteria as it currently stands. Therefore, we invite you to submit a revised version of the manuscript that addresses the last  points raised during the review process.

We look forward to receiving your revised manuscript.

Kind regards,

Ray Borrow, Ph.D., FRCPath

Academic Editor

PLOS ONE

Journal Requirements:

Reviewers' comments:

Reviewer's Responses to Questions

**Comments to the Author**

1. If the authors have adequately addressed your comments raised in a previous round of review and you feel that this manuscript is now acceptable for publication, you may indicate that here to bypass the “Comments to the Author” section, enter your conflict of interest statement in the “Confidential to Editor” section, and submit your "Accept" recommendation.

Reviewer #1: (No Response)

Reviewer #2: All comments have been addressed

Reviewer #3: All comments have been addressed

2. Is the manuscript technically sound, and do the data support the conclusions?

Reviewer #1: (No Response)

Reviewer #2: Yes

Reviewer #3: Yes

3. Has the statistical analysis been performed appropriately and rigorously? 

Reviewer #1: (No Response)

Reviewer #2: Yes

Reviewer #3: Yes

4. Have the authors made all data underlying the findings in their manuscript fully available?

Reviewer #1: (No Response)

Reviewer #2: Yes

Reviewer #3: Yes

5. Is the manuscript presented in an intelligible fashion and written in standard English?

Reviewer #1: (No Response)

Reviewer #2: Yes

Reviewer #3: Yes

6. Review Comments to the Author

Reviewer #1: “Placental malaria prevalence”- is this a binary (presence/absence) variable or a continuous variable?

Cox regression: “total IgG”- is the sum of IgG across IgG1, IgG2, IgG3 and IgG4? There are many IgG variables in the model and there may be collinearity. My suggestion is to include only IgG specific to one antigen at a time as a predictor in the cox regression while adjusting for other non-IgG covariates. The p-values of the IgG variables from all the different cox models should be adjusted for multiple testing.

Why wasn’t mother’s economic status included as a covariate in the model?

Cox regressions adjusting for potential confounders are more informative than Kaplan-Meir survival analysis. The section “Probability of survival to “probability of fewer occurrence of malaria”” seems to be superfluous.

The section “Effects of Economic status on birth outcomes and risk of malaria incidence”- the only association shown is with birth weight, which is not relevant to the main topic of this paper. If there is no association with malaria incidence or IgG levels, the entire section can go. Please go through the paper carefully and remove any language related to this analysis.

Reviewer #2: The authors have done an excellent job. I read with great interest the responses to the various comments from reviewers.

A minor error found in the “laboratory procedures”: 50μl of a secondary antibody specific for an IgG subtype (IgG1, IgG2, IgG2, IgG3 and IgG4)...IgG2 is repeated twice.

Manuscript can be accepted

Reviewer #3: Authors should attempt to further clean up their figures, and will need to submit publication-quality for the final version.

7. PLOS authors have the option to publish the peer review history of their article (what does this mean?). If published, this will include your full peer review and any attached files.

Reviewer #1: No

Reviewer #2: No

Reviewer #3: No

---

## [Author Response · Author response to Decision Letter 3]

22 Sep 2022

Reviewer #1, Qn 1: “Placental malaria prevalence”- is this a binary (presence/absence) variable or a continuous variable?

Yes, placental malaria is a binary variable and was measured as a Yes or No. This has been clearly indicated in the revised Manuscript.

Reviewer # 1, Qn 2: Cox regression: “total IgG”- is the sum of IgG across IgG1, IgG2, IgG3 and IgG4? There are many IgG variables in the model and there may be collinearity. My suggestion is to include only IgG specific to one antigen at a time as a predictor in the cox regression while adjusting for other non-IgG covariates. The p-values of the IgG variables from all the different cox models should be adjusted for multiple testing.

Response: Yes, we agree to the reviewer’s suggestion, and we included only IgG specific to one antigen at a time in building the Cox regression model while adjusting for non-IgG covariates at each time. The Cox regression model was built using back ward elimination method. For each IgG, the levels were grouped into percentiles, that is 25th, 50th and 75th . The effects of different levels of the IgG (25th,50th and 75th percentiles) on the incidence of malaria among study participants were assessed. The levels of the IgG at the 75th percentile was found to consistently have higher Hazard ratios and significant confidence intervals. Therefore, the levels of the IgG at the 75th percentile were used in building the Cox regression model. The P-values of the IgG variables in the Cox regression models were adjusted for Multiple testing. The non-IgG covariates included birth weight (<2500 grams, >2500grams), maternal malaria incidence (No malaria detected, and malaria detected), Maternal wealth index (poorest, middle, and least poor), gravidity category (1-3 and 4+). Cofounding was assessed in the Cox regression model and any variable that had the percentage difference of more than 10% between the crude and adjusted Hazard ratios was considered a cofounder. In the model building, the covariates that had p value of ≤ 0.2 in the bivariate analysis were included in the model. 

Reviewer #1,Qn 3:Why wasn’t mother’s economic status included as a covariate in the model?

Response: Yes, we agree with the reviewer’s suggestion, in the revised manuscript, we have included Mother’s wealth index (Poorest, middle and least poor)

Reviewer #1, Qn4, Cox regressions adjusting for potential confounders are more informative than Kaplan-Meir survival analysis. The section “Probability of survival to “probability of fewer occurrence of malaria”” seems to be superfluous.

Response: Yes, we agree with the reviewer’s suggestion and in the revised manuscript, we developed a Cox regression model and adjusted for multiple testing in the IgG covariates and for cofounding in all other variables including the mother’s characteristics. . 

Reviewer #1, Qn5: The section “Effects of Economic status on birth outcomes and risk of malaria incidence”- the only association shown is with birth weight, which is not relevant to the main topic of this paper. If there is no association with malaria incidence or IgG levels, the entire section can go. Please go through the paper carefully and remove any language related to this analysis.

Response: Guidance well noted; section on “Effects of Economic status on birth outcomes” has been removed as advised in the revised manuscript. We included Mother’s wealth category in the Cox regression model and assessed if it cofounds the estimation of malaria incidence amongst study participants

Reviewer #2: The authors have done an excellent job. I read with great interest the responses to the various comments from reviewers.

A minor error found in the “laboratory procedures”: 50μl of a secondary antibody specific for an IgG subtype (IgG1, IgG2, IgG2, IgG3 and IgG4)...IgG2 is repeated twice.

Response: Thank you so much for your keen observation; IgG2 was indeed repeated. This has been corrected in the revised manuscript.

Reviewer #3: Authors should attempt to further clean up their figures and will need to submit publication-quality for the final version.

Response: Yes, we agree with the reviewer’s comments, and we have improved the quality of the figures in the revised manuscript.

---

## [Editor Report · Decision Letter 4]

4 Nov 2022

Effects of anti-malarial prophylaxes on maternal transfer of Immunogloublin-G (IgG) and  association to Immunity against Plasmodium falciparum infections among Children in a Ugandan birth Cohort

PONE-D-21-38887R4

Dear Dr. Okek,

We’re pleased to inform you that your manuscript has been judged scientifically suitable for publication and will be formally accepted for publication once it meets all outstanding technical requirements.

Kind regards,

Ray Borrow, Ph.D., FRCPath

Academic Editor

PLOS ONE
---

## [Editor Report · Acceptance letter]

17 Jan 2023

PONE-D-21-38887R4 

Effects of anti-malarial prophylaxes on maternal transfer of Immunoglobulin-G (IgG) and association to Immunity against *Plasmodium falciparum* infections among Children in a Ugandan birth Cohort 

Dear Dr. Okek:

I'm pleased to inform you that your manuscript has been deemed suitable for publication in PLOS ONE. Congratulations! Your manuscript is now with our production department. 

Kind regards, 

on behalf of

Prof. Ray Borrow 

Academic Editor

PLOS ONE